# 16S rRNA Gene Sequencing Revealed Changes in Gut Microbiota Composition during Pregnancy and Lactation in Mice Model

**DOI:** 10.3390/vetsci9040169

**Published:** 2022-04-01

**Authors:** Mengmeng Guo, Xi Cao, Ke Zhang, Menghao Pan, Yujiang Wu, Suo Langda, Yuxin Yang, Yulin Chen, Ba Gui, Baohua Ma

**Affiliations:** 1Key Laboratory of Animal Biotechnology, Ministry of Agriculture, College of Veterinary Medicine, Northwest A&F University, Yanglin 712100, China; guomengmeng@nwafu.edu.cn (M.G.); panmenghao@nwafu.edu.cn (M.P.); 2Key Laboratory of Animal Genetics, Breeding and Reproduction of Shaanxi Province, College of Animal Science and Technology, Northwest A&F University, Yanglin 712100, China; xndkcx@nwafu.edu.cn (X.C.); kezhang@nwafu.edu.cn (K.Z.); yxyang@nwafu.edu.cn (Y.Y.); chenyulin@nwafu.edu.cn (Y.C.); 3Institute of Animal Sciences, Tibet Academy of Agricultural and Animal Husbandry Sciences, Lhasa 850009, China; wuyujiang_1979@163.com (Y.W.); sonaada10@163.com (S.L.)

**Keywords:** gut microbiota, pregnancy, lactation, reproductive hormones, metabolites

## Abstract

The gut microbiota play a vital role in regulating endocrine-mediated metabolism, immunity, and energy metabolism. However, little is known about the gut microbiota and metabolite composition and development throughout pregnancy and lactation. Here, we used amplicon sequencing to analyze the gut microbiota composition of 35 female mice in five stages of pregnancy and lactation, namely, non-receptive (NR) stages, sexually-receptive (SR) stages, at day 15 of pregnancy (Pre_D15), at the day of birth (Del), and at day 10 of lactation (Lac_D10). The results revealed that the α diversity of gut microbiota was significantly increased during pregnancy and lactation. In addition, the principal coordinate analysis (PCoA) conducted on the amplicon sequence variants’ (ASVs’) distribution of the 16S rRNA amplicons indicated that the microbiota composition was significantly different among the five groups. Based on a random forest analysis, *Oscillospira*, *Dehalobacterium,* and *Alistipes* were the most important microbiota. The abundance of *Allobaculum*, *Oscillospira,* and *Ruminococcus* were negatively correlated with the serum progesterone concentration, while the abundance of *Oscillospira* was positively correlated with the propionate and valerate concentration in the caecal contents. Moreover, the concentration of acetate and propionate in the Del and Lac_D10 stages was significantly lower than in the SR and Pre_D15 stages. Our findings indicate that some gut microbes and metabolites vary considerably at the different stages of pregnancy and during lactation stages, which can potentially be used as microbial biomarkers. These results provide information on the potential use of the identified microbes as probiotics to maintain a healthy pregnancy and lactation.

## 1. Introduction

The gut microbiota play a critical role in various physiological processes, including hormone-mediated metabolism [1,2], immune system development [3], and energy metabolism [4]. During the first and early second trimesters of pregnancy, the maternal body exhibits strong inflammatory responses [5]. However, the placenta has certain anti-inflammatory properties, which protect the fetus from expulsion [6]. Moreover, the maternal intestinal epithelium also exhibits a characteristic mild inflammation following the expression of pro-inflammatory factors, which increases with pregnancy length [6,7]. As the maternal pro-inflammatory activity and the immune levels change during pregnancy, the reproductive hormone levels also exhibit a drastic change, which influences the gut bacterial diversity and function [8]. Among them, the levels of estrogen and progesterone modulate the gut bacteria metabolism, growth, and virulence of pathogenic bacteria, indirectly affecting the composition of the maternal gut microbiota [5,9]. In addition, this poses a risk for preterm delivery or stillbirth [10]. At the same time, the drastic changes in the maternal reproductive hormones before and after childbirth also affect the contractility and transport in the intestine, which affect nutrient absorption and energy metabolism, and promote maternal weight gain [11]. Therefore, exploring the changes in gut microbiota and metabolites during and after pregnancy could help understand their contribution to weight gain and host physiology changes.

Gestational diabetes mellitus, energy metabolism, and immunologic adaptations are associated with a change in the gut microbiota composition in humans [7,12], pigs [13], and ruminants [4] during pregnancy. For example, in mice and humans, the estrogen levels affected the gut microbiota [14]. The gut microbiota secretes β-glucuronidase (an enzyme deconjugating estrogen), which binds to the estrogen receptor, affecting the estrogen level in the host [15], and subsequently affecting the downstream physiological processes [16]. In women and mice, the gut microbial composition in late pregnancy undergoes tremendous changes, including an increased abundance of *Bifidobacterium* [17]. However, the systemic changes of gut microbiota and metabolite composition from the beginning of pregnancy to lactation have not been studied. Identifying the microbial biomarkers and understanding the change process of short-chain fatty acids (SCFAs) at each stage of pregnancy are essential for healthy pregnancy intervention using targeted probiotics. 

In this study, the 16S rRNA gene sequencing and gas chromatography techniques were used to explore gut microbiota and SCFAs’ composition changes to identify the microbial biomarkers inducing metabolic changes during pregnancy and lactation. The findings revealed that the gut microbial community and SCFAs’ composition are profoundly altered from non-pregnancy to pregnancy and lactation, which impacts the host reproductive hormone levels that are beneficial during pregnancy and lactation. This study can help to identify new target probiotics that could promote optimal gut health and hormone levels in pregnancy and lactation stages in animals.

## 2. Materials and Methods

### 2.1. Animals, Experimental Design, and Sample Collection

The experimental procedures were approved by the Ethics Committee of Northwest A&F University (approval number: 20200515–010). A total of 60 specific pathogen-free (SPF) Kunming (KM) female mice and 30 KM male mice aged eight weeks were obtained from the Laboratory Animal Center of Fourth Military Medical University (Xi’an, China). All the mice were housed in cages in a pathogen-free animal facility at the College of Veterinary Medicine in Northwest A&F University (Yangling, China). Using the vaginal smear method, the mice were continuously observed for two estruses to determine the time of each estrus cycle. On the afternoon of the day before estrus, male and female mice were placed in the same cage for mating at a ratio of 1:2. On the morning of the second day, a vaginal suppository was used to determine whether fertilization had occurred, and this day was set as the first day of pregnancy. Finally, 35 mice were determined for follow-up experiments. All mice were kept in a strictly controlled environment to maintain a normal circadian rhythm, the mice were housed individually at room temperature (23 ± 1 °C) and under a 12 h light/12 h dark cycle, and allowed free access to a standard rodent diet (Xietong, Jiangsu) and water. The diet nutrient composition is provided in the Appendix A. The diet of all mice did not change during the whole trial period. 

The caecal contents and mice serum were collected during the non-receptive (NR) stages (diestrus and proestrus; *n* = 7), sexually-receptive (SR) stages (estrus and metestrus; *n* = 7), (Chari et al. 2020), at day 15 of pregnancy (Pre_D15; *n* = 7), at the day of birth (Del; *n* = 7), and day 10 of lactation (Lac_D10; *n* = 7) (Figure 1A). Blood samples (2 mL) were collected from the facial vein and centrifuged at 3000 rpm for 10 min at 4 °C to obtain the serum. The obtained serum was divided into two tubes and stored at −20 °C prior to the assay. After serum collection, the mice were euthanized by intraperitoneal injection of 200 mg/kg pentobarbital and caecal contents were collected. All caecal content samples were immediately snap-frozen in liquid nitrogen and stored at −80 °C for downstream investigations.

### 2.2. Identification of Mouse Estrus Cycle Stage 

The 35 mice were assigned into two groups (NR and SR) based on the stages of the estrus cycle determined through cytological evaluation. The estrus cycle was identified by standard procedures [18]. Before any tests, the estrus cycle for each mouse was ensured to be normal. Mice in the SR group were mated for subsequent testing (Figure 1B). Vaginal flushing fluid samples were collected for making slides, and stained with hematoxylin and eosin staining (H&E; hematoxylin staining for 1 min and eosin staining for 50 s), then processed, dried, subjected to microscopic examination and micrography [18].

### 2.3. Sex Hormones Measurements

The levels of serum estradiol (E_2_) were measured using the HY-10029 IRMA KIT (Huaying, Beijing, China) and the levels of serum progesterone (P) were measured using the HY-10028 RIA KIT (Huaying, Beijing, China) in accordance with the manufacturer’s instructions.

### 2.4. Analysis of Cecum Short-Chain Fatty Acids (SCFAs)

For each caecal content sample, 0.03 g caecal content was weighed into a 2 mL centrifuge tube, and 400 μL ultrapure water was added. The sample was shaken until homogenized and incubated for 30 min. The samples were centrifuged at 10,000 rpm for 10 min at 4 °C to collect 300 μL of the supernatants. To the supernatants, 30 μL of metaphosphoric acid was added, homogenized by vortexing, and incubated for 3–4 h at 4 °C. Centrifugation was then performed at 13,500 rpm for 15 min at 4 °C to separate the protein and impurities. The supernatant was aspirated, and trans-crotonic acid in a volume equal to the supernatant was added. The mixer was incubated for 20 min, after which a 0.22 μm water phase filter membrane was used to separate the caecal content from the supernatant. The supernatants were stored in 2 mL screw-cap vials. The SCFAs in the caecal content were analyzed by gas chromatography (Agilent Technologies 7820A GC system, Santa Clara, CA, USA) using a 30 mm, 0.25 mm, and 0.33 μm fused silica column (AE-FFAP; ATECH Technologies Co., Ltd., Lanzhou, China) [19].

### 2.5. DNA Extraction, PCR Amplification, and 16S rRNA Gene Sequencing

Microbial DNA was extracted from the caecal contents using the Tiangen DNA Stool Mini Kit (DP328, Tiangen, Beijing, China) following the manufacturer’s instructions. The extracted DNA was then quantified using the Nanodrop 2000 UV-VI spectrophotometer (Thermo Scientific, Wilmington, NC, USA). The primers 338F (5′-ACTCCTACGGGAGGCAGCAG-3′) and 806R (5′-GGACTACHVGGGTWTCTAAT-3′) on a GeneAmp PCR system 9700 thermal cycler (Applied Biosystems, Foster City, CA, USA) were used to amplify the V3–V4 region of the DNA [20]. Amplification was performed under the following conditions: initial denaturation at 98 °C for 2 min, denaturation at 98 °C for 15 s, annealing at 55 °C for 30 s, extension at 72 °C for 30 s, repeated for 30 cycles, and final extension at 72 °C for 5 min. The amplicons were then excised from the 1.5% agarose gel and purified using the QIAquick Gel Extraction Kit (28706, Qiagen, Germany). Purified amplicons were pooled in equimolar ratios and subjected to paired-end sequencing (2 × 300 bp) on an Illumina MiSeq platform (Illumina, San Diego, California, CA, USA) according to the standard protocols by Personal Biotechnology Co., Ltd., Shanghai, China. 

### 2.6. 16S rRNA Gene Sequencing Data Analysis

The FASTQ data were imported into the QIIME2 (version 2019.4, https://qiime2.org/, accessed on 8 October 2020) platform [21], and were processed by the GenesCloud platform (https://www.genescloud.cn/home, accessed on 10 October 2020). Briefly, after demultiplexing, the resulting sequences were merged with FLASH (version 1.2.11, https://www.flash.cn/, accessed on 15 October 2020) [22] and quality filtered with FASTP (version 0.19.6, https://github.com/OpenGene/fastp, accessed on 17 October 2020) [23]. DADA2 plugin was used to quality filter, denoise, merge and remove chimera [21]. The reads from each sample were rarefied to 60,000, and average Good’s coverage achieved 99.51%. Taxonomic assignment of amplicon sequence variants (ASVs) was performed using the naive Bayes consensus taxonomy classifier (Bokulich et al. 2018) implemented in QIIME2 and against the reference sequences in the Greengenes database (Release 13.8, http://greengenes.secondgenome.com, accessed on 22 October 2020) [24]. Sequences representing chloroplasts and mitochondria were filtered out. A total of 5.16% of the clean data could not be detail annotated at the phylum level, and the non-annotated data were not included in the subsequent analysis. This Greengenes database was found to provide good quality and comprehensive profile of gut microbiome [25,26]. To the best of our knowledge, the Greengenes database has not been updated since 2013. Therefore, appropriate databases should be carefully selected for using in subsequent analyses. Mothur-1.30 was used to calculate the Alpha diversity index under different random samples. β-diversity (PCoA based on Bray–Curtis dissimilarities) was calculated using the R-3.3.1 (vegan) package, https://r-pkgs.org, accessed on 5 November 2020. For PCoA analysis, we used the ANOSIM function in the vegan package of R including different independent variables with default settings (999 permutations). 

### 2.7. Statistical Analysis 

Statistical analyses were performed using SPSS Statistics V21.0 software (https://www.ibm.com, accessed on 5 November 2020). The normally distributed continuous variables were analyzed using an unpaired Student’s *t*-test and the one-way ANOVA with Tukey’s post hoc test. Treatment differences with *p* < 0.05 were considered statistically significant. The Shannon diversity, Chao1 diversity, and Phylogenetic diversity were calculated by QIIME2 and compared using the Kruskal–Wallis test. The differences in microbiota composition among treatments were visualized using the principal coordinate analysis (PCoA) plots based on the Jaccard index distance metric. A combined LEfSe and indicator species analysis in the “indicspecies” package in R was then performed to identify the core bacterial ASVs predominant in NR, SR, Pre_D15, Del, and Lac_D10 groups, and the threshold on the logarithmic, linear discriminant analysis (LDA) score was set to 2.5. Heatmap analysis using R packages (V3.5.2) was used to build based on the top 50 genera, https://r-pkgs.org, accessed on 5 November 2020. Random forest analyses were performed using the “randomForest” v.4.6-14 package in R [27]. The significant difference of microbial taxa was tested using non-parametric Kruskal–Wallis tests. The Spearman rank correlation test between the SCFAs and the bacterial taxa was performed at the genus level using R packages (V3.5.2). 

## 3. Results

### 3.1. Weight and SERUM Sex Hormones Changes during Pregnancy and Lactation

The body weight of mice prior to estrus identification was defined as initial body weight, and there was no significant difference between groups (Figure 2A). The mice in the Pre_D15, Lac_D10, and Del groups recorded a significantly higher final weight than those in the NR group at *p* < 0.001 (Figure 2A). In addition, the level of estrogen in the serum for the SR group was significantly higher than in the Pre_D15 and Lac_D10 groups at *p* < 0.001 (Figure 2B). In contrast, the SR group recorded a significantly low level of progesterone (P) in the serum when compared to the NR, Pre_D15, and Lac_D10 groups at *p* < 0.001 (Figure 2C). However, at *p* < 0.001, the Pre_D15 and Lac_D10 groups recorded a significantly higher P than the Del group (Figure 2C).

### 3.2. Microbial Community Profiles during Pregnancy and Lactation in Mice 

A total of 2,653,875 high-quality reads from 35 samples with an average of 60,000 clean reads per sample were obtained after quality control. The rarefaction curves became increasingly smooth with the increase in sequencing quantity, implying the data quality was reliable (Appendix A). Among the five groups, the NR group had the lowest Shannon and Simpson index (Figure 2D, Appendix A), and Chao 1 observed species and Good’s coverage were not significantly different in the five groups (Appendix A). The Shannon index was the highest in the pregnancy stage and decreased as the lactation stage progressed (Figure 2D). The PCoA conducted on the ASVs’ distribution of the 16S rRNA data revealed that the gut microbiota is significantly different among the five groups using an analysis of similarity based on the Jaccard index (ANOSIM, R = 0.292, *p* = 0.001; Figure 2E). However, the number of ASVs in each group was not significantly different (Appendix A). Interestingly, the gut microbiota in the NR, Del, and Lac_D10 groups formed three distinct clusters based on the Jaccard index (ANOSIM, R = 0.272, *p* = 0.001; Figure 2E). 

### 3.3. Comparison of the Gut Microbiota during Pregnancy and Lactation

Phylum-level classification of the mouse gut microbiota revealed a stable composition in the different groups. Additionally, a total of eight phyla and twenty-five genera were observed in all samples. The majority of the bacterial phyla identified in the caecal content samples were encompassed by Firmicutes (75%), Bacteroidetes (15%), Proteobacteria (5%), and Actinobacteria (2%), as shown in Figure 2F. Furthermore, Firmicutes and Bacteroides accounted for more than 90% of the total microbiota (Figure 2F). At phylum-level, from NR to Lac_D10, the relative abundances of Bacteroidetes increased on average (*p* = 0.002), the relative abundances of Proteobacteria increased from Del to Lac_D10 (*p* = 0.029; Figure 2G), and the relative abundance of Firmicutes decreased from NR to SR (*p <* 0.009; Figure 2G). The ratio of Firmicutes/Bacteroidetes (Fir/Bac) of the NR group was significantly higher than that of other groups (Figure 2H).

However, the dominant genera in the different groups were considerably varied. In the Lac_D10 group, the dominant genera were *Desulfovibrio*, *Bacteroides*, *Helicobacter*, *Alistipes*, *Turicibacter*, *Sutterella*, *Psychrobacter*, *Acinetobacter*, *Anaeroplasma*, *Odoribacter*, *Rikenella*, *Jeotgalicoccus*, *Anaerostipes*, *Mycoplasma,* and *Candidatus Arthromitus*, while *Streptococcus*, *Coprococcus*, and *Bilophila* dominated the Del group. In the SR group, *Clostridium*, *Parabacteroides*, *Roseburia,* and *Prevotella* were the dominant genera. The dominant genera in the Pre_D15 group were *Butyricicoccus*, *Alcaligenes*, *AF12*, *Paracoccus*, *Corynebacterium*, *SMB53*, *Facklamia*, and *Brachybacterium Salinicoccus; Yaniella*, *Lactobacillus*, *Virgibacillus*, *Allobaculum*, *Gemella*, *Adlercreutzia*, and *Sporosarcina* were dominant in *the* NR group (Figure 3A). At genus level, the relative abundances of *Lactobacillus* and *Bacteroides* were the lowest in the NR group (*p <* 0.05; Appendix A). The abundance of *Rikenella* was significantly increased from NR to Lac_D10 (*p* = 0.019; Appendix A). Intriguingly, *Clostridium* and *Sphingomonas* were not detected in the NR and SR groups (Appendix A). *Oscillospira* and *Ruminococcus* were enriched in the SR, Pre_D15, and Del groups (*p <* 0.05; Appendix A).

To analyze the differences in bacteria in different physiological period groups. We compared the microbial abundance between other physiological groups and the NR group. We found that the significantly enriched bacterial orders were *Bacteroidales*, *Clostridiales*, and *Desulfovibrionales* in the SR group, *Clostridiales* in the Pre_D15 group, *Bacteroidales* and *Clostridiales* in the Del group, and *Coriobacteriales*, *Burkholderiales*, *Bacteroidales*, *Bacillales*, *Clostridiales*, *Lactobacillales*, and *Anaeroplasmatates* in the Lac_D10 group (Figure 3B). Based on the LEfSe analysis, *Bacteroidia*, *S24-7*, *Oscillospira*, *Ochrobactrum*, and *Dehalobacterium* were enriched in the SR group; *Lachnospiraceae*, *Ruminococcus*, and *Coprococcus* in the Del group, and *Rikenellaceae*, *Rikenella*, and *Proteobacteria* in the Lac_D10 group (All-against-all; LDA score (log 10) > 2.5; Figure 3C). In addition, the random forest analysis revealed that *Oscillospira*, *Dehalobacterium,* and *Alistipes* were the most important genera in the model (Figure 3D). The Spearman’s rho coefficients of the top 30 genera further revealed that *Allobaculum*, *Oscillospira*, unidentified *Mogibacteriaceae*, unidentified *Lachnospiraceae*, *Ruminococcus,* and unclassified *Clostridiales* negatively correlated with p-levels. Moreover, the unidentified *F16* abundance positively correlated with P (Figure 4). These results indicate that the caecal microbiota composition and abundance is greatly altered during pregnancy and lactation.

### 3.4. Fecal SCFAs Concentrations during Pregnancy and Lactation in Mice

Gut microbiota ferments carbohydrates to produce SCFAs, including acetate, propionate, isobutyrate, butyrate, isovalerate, and valerate. The assessment of bacterial community impacts on the synthesis of SCFAs revealed no significant alteration in the concentrations of acetate, propionate, and total SCFAs in the NR, Pre_D15, Del, and Lac_D10 groups (Figure 5). However, the concentrations of acetate, propionate, and SCFAs in the Del and Lac-D10 groups were significantly lower than in the SR group. In contrast, the butyrate concentration remained unaltered among the five groups (Figure 5).

### 3.5. Correlation Analysis of Caecal Microbiota Composition and SCFAs

Spearman rank correlation analysis between Top 50 taxa at genus level and SCFAs revealed that *Oscillospira* was positively correlated with propionate and valerate, and *U-Lachnospiraceae* and *Rikenella* were positively correlated with valerate at *p* < 0.05 (Figure 6). In addition, isobutyrate positively correlated with *U-Lachnospiraceae, Rikenella, U-Clostridiaceae, U-staphylococcaceae, parabacteroides, U-Clostridiales, Desulfovibrio*, *and* unidentified_*S24-7*, while butyrate positively correlated with *U-Mogibacteriaceae* at *p* < 0.05. However, *Allobaculum* was negatively correlated with isovalerate and *Adlercreutzia* with butyrate at *p* < 0.05 (Figure 6).

## 4. Discussion

From the beginning of pregnancy, the mothers’ body undergoes a series of dynamic changes in nutritional metabolism, immune function, and the endocrine system to meet the nutritional and immune needs of fetal development [28,29,30]. In this study, we explored the impact of pregnancy and lactation on gut microbiota composition in mice under highly diet-controlled conditions. Our study found that some of the gut microbes varied considerably in the gestation or lactation stages. As mentioned earlier, Firmicutes and Bacteroidetes are dominant regardless of the stage of reproduction [31]. The relative abundance of Firmicutes and Bacteroides is related to human and mouse body fat content [32,33,34]. A decreased ratio of major Fir/Bac ratio can prevent obesity in mice’s dietary and genetic obesity models [35]. Thus, their decrease subsequently terminates fat deposition in the second trimester, it is speculated that after pregnancy begins, the body is more inclined to synthesize breast milk using pre-pregnancy stores of nutrients and energy rather than storing fat [34,36]. 

The alpha diversity increased during pregnancy possibly to meet the maternal metabolic requirements [37]. In the current study, the Shannon index was lower on the Lac_D10 group. Chatelier et al. showed a positive correlation between low gut microbiota diversity in negative metabolic conditions, including insulin resistance and gut inflammatory phenotype [38]. Increased insulin resistance in the mother leads to increased insulin-mediated glucose and fatty acid levels, providing more energy sources for offspring [39]. *Lactobacillus* found in the gut and utilized as probiotics have also been associated with obesity [40,41,42]. Probiotics have been found to prevent obesity-related diseases, such as hypertension, hyperglycemia, and dyslipidemia. Moreover, *Lactobacillus fermentum* CECT5716 supplements during pregnancy and lactation stages improve the mother’s health, especially in the intestines, which boosts the mother and offspring’s immunity [43]. In our study, *Lactobacillus* decreased during pregnancy and lactation compared to the NR period, which may be related to the decreased immunity of the mother and fetus during pregnancy.

A previous study suggested that low gut microbiota richness in overweight pregnant women is associated with increased gut permeability [44]. The relative abundance of *Oscillospira* is associated with the body’s inflammatory responses and gut barrier function [45]. Our study found that *Oscillospira* was lowest in the NR group, higher in pregnancy, and lower in lactation. *Oscillospira* is involved in lipolysis, glucose synthesis, and hormone secretion. A previous study also found that *Oscillospira* were positively correlated with plasma E_2_ levels in the pregnancy and lactation stage in a pig model [46]. Moreover, in menopausal women with reduced estrogen levels and bone density, the low bone density women had lower species diversity, a higher abundance of γ-Proteobacteria, and a lower intake of lycopene compared with the normal women, and Lycopene intake positively correlated with *Oscillospira* genus abundance [47]. Previous studies found that a high abundance of *Oscillospira* was detected in the gut of patients with chronic kidney disease [48]. In addition, the abundance of *Oscillospira* was positively associated with Parkinson’s disease, and central neurological and degenerative disorders [49]. In addition, a series of evidence shows that the abundance of *Oscillospira* is related to the concentration of serum E_2_. Correlation analysis revealed a significant positive correlation between *Oscillospira* and the production of propionate and valerate. However, propionate intake may lead to insulin resistance [50]. In this study, we found that the relative abundance of *Oscillospira* in pregnancy was higher than in non-pregnancy, which may be related to the susceptibility of pregnant women to gestational diabetes [51]. Currently, *Oscillospira*’s correlation has been explored only in metagenomics data related to the gut microbiota, and the pure cultures and identification of this strain have not been obtained [52]. Therefore, the causal biology functions of *Oscillospira* strains have not been conclusively established in mice and human models. The current research of the *Oscillospira* genus is mainly reflected in the change in abundance in different animal models [53]. Taken together, *Oscillospira* exhibits beneficial microbial characteristics and has shown a positive biology function in different disease models, such as obesity and chronic inflammation [22,54]. Therefore, we should actively research the function of *Oscillospira* strains, and develop the potential probiotics to regulate the gut health of disease.

Moreover, during the late gestation period, a decrease in propionic acid leads to increased adipose decomposition, which stimulates the milk synthesis preparation [55,56]. The gut microbiota mainly influence their host’s metabolic level and processes through SCFAs’ production [57]. Intestinal microbiota dysbiosis may be one of the pathogenesis of type 1 diabetes. A previous study found that the key characteristics of the disease were negatively correlated with the metabolites of acetate and butyrate in blood and feces in the non-obese diabetic mouse model [58]. In this study, the results showed that the acetate levels were higher in early pregnancy, again speculating on the causes of diabetes during pregnancy. The SCFAs also maintain intestinal micro-ecology systems by reducing the pH [59]. Butyrate maintains human intestinal health, in addition to providing energy to the colonic mucosa and host cell gene expression, differentiation, and apoptosis of important regulators of inflammation [59]. The change in gut microbiota composition can affect the concentration of essential SCFAs [60]. SCFAs act as a bridge between microbes and the body to regulate metabolic balance. There are a number of changes in the mother throughout the reproductive period in order to provide sufficient energy and nutrition for the offspring. However, SCFAs also increase the risk of metabolic disorders. The genus of *Oscillospira* was negatively correlated with P during pregnancy. We hypothesized that *Oscillospira* can regulate host energy metabolism during pregnancy. Further investigation is required to verify the function of *Oscillospira* Sp. through in vitro cultivation and the use of animal models.

## 5. Conclusions

The present study found that the maternal gut microbiota composition and metabolic profiles drastically change over pregnancy and during lactation. The representative changes include an increase in Proteobacteria and *Rikenella* during the lactation stage. Moreover, animals have a very specific gut microbial composition during gestation and lactation. The enrichment of *Oscillospira* during pregnancy is negatively correlated with progesterone concentration and positively correlated with the concentration of propionate and valerate. This study identified that significant pregnancy-related microbes offer a novel view, and provided a theoretical basis for the use of probiotics to interfere with healthy pregnancies and lactation in animals. 

## Figures and Tables

**Figure 1 vetsci-09-00169-f001:**
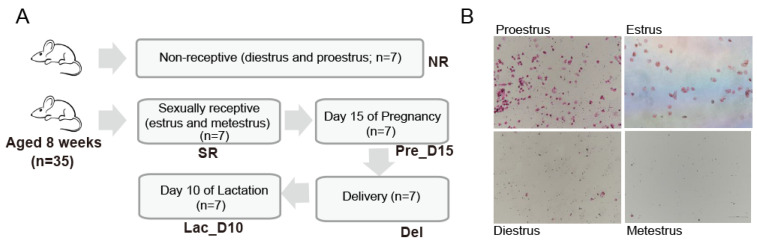
Stage identification of mouse estrus cycle. (**A**) Sampling scheme. (**B**) Staging of the mouse estrus cycle, including four distinct stages (leukocytes, red represent nucleated epithelial cells, blue represent cornified epithelial cells, yellow arrow). Sexually receptive including metestrus and estrus (SR; *n* = 7), and non-receptive including diestrus and proestrus (NR; *n* = 7).

**Figure 2 vetsci-09-00169-f002:**
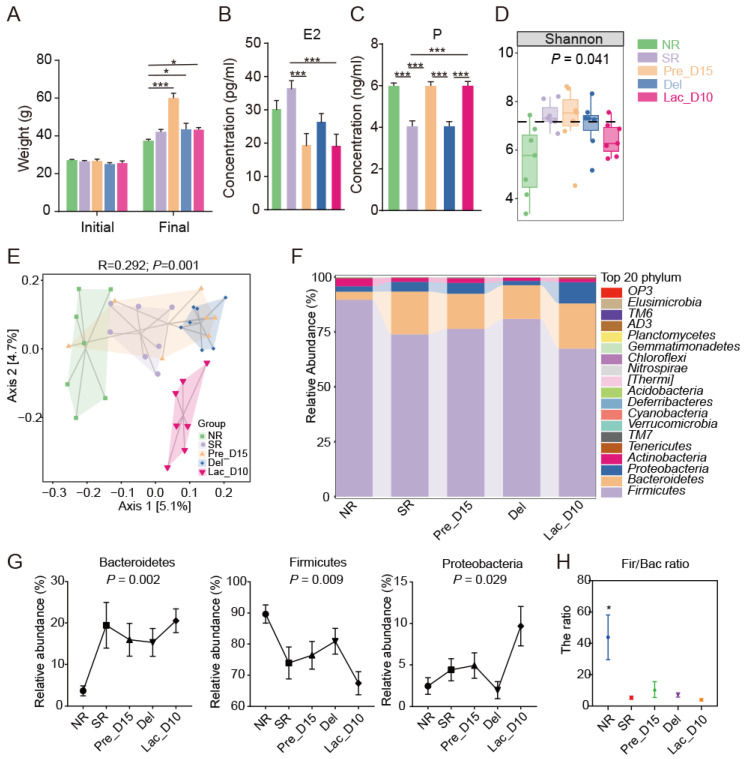
The reproductive hormone levels and gut microbiota composition in mice. (**A**) Mice body weights at different reproductive stages. (**B**) Levels of estrogen in serum. (**C**) Levels of progesterone in serum. (**D**) Alpha diversity (Shannon evenness) of microbial communities in five groups. (**E**) PCoA plot based on the weighted UniFrac distance matrix. (**F**) The caecal microbiota composition at the phylum level in five groups. (**G**) Gut microbiota changes in core phylum among five groups. The non-parametric Kruskal–Wallis tests was used to analysis the difference. (**H**) The ratio of Fir/Bac in the five groups. The results are presented as mean ± SEM * *p* < 0.05; *** *p* < 0.001.

**Figure 3 vetsci-09-00169-f003:**
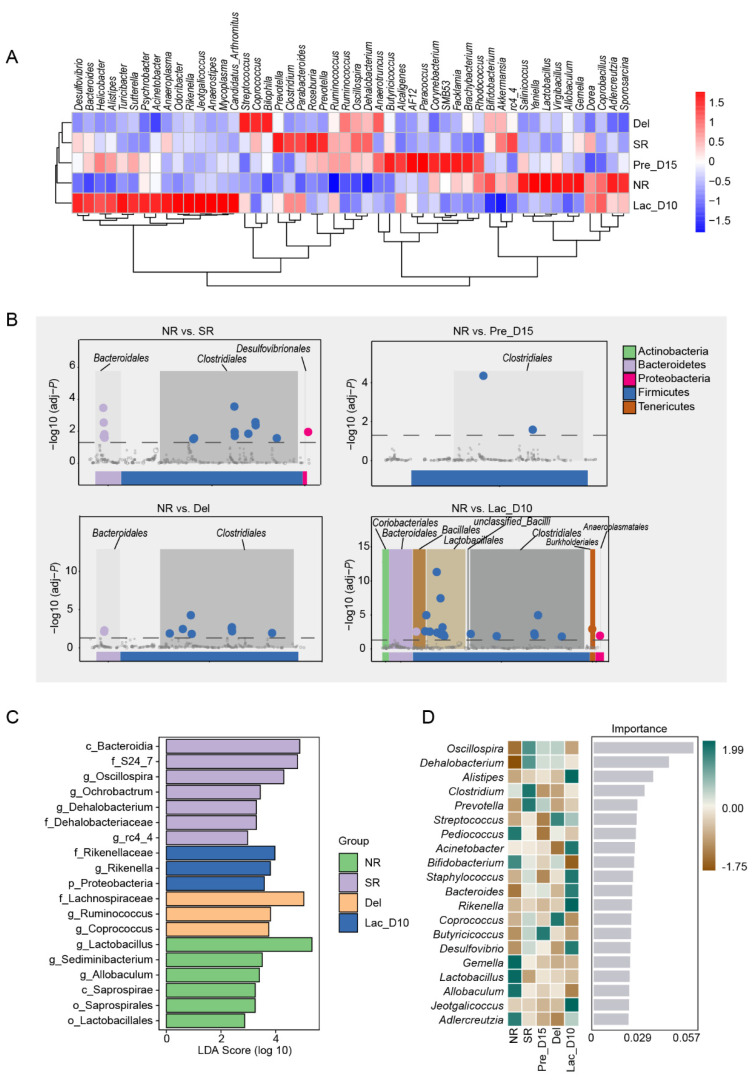
Composition of caecal content microbiota of mice during different stages of the reproductive cycle. (**A**) The genus-level abundance of caecal microbiota in five groups. (**B**) Manhattan plots depicting the distribution of caecal bacteria during different groups. The circles represent microbiota at the order level; the bar chart represents bacteria at the phylum level. (**C**) The LDA scores for the differential taxa in caecal samples during the different reproductive stages. (**D**) The genus-level abundance of caecal microbiota in the different genera. The first column represents the microbiota at the genus level. The second to last column represents the abundance of the corresponding genus in each group.

**Figure 4 vetsci-09-00169-f004:**
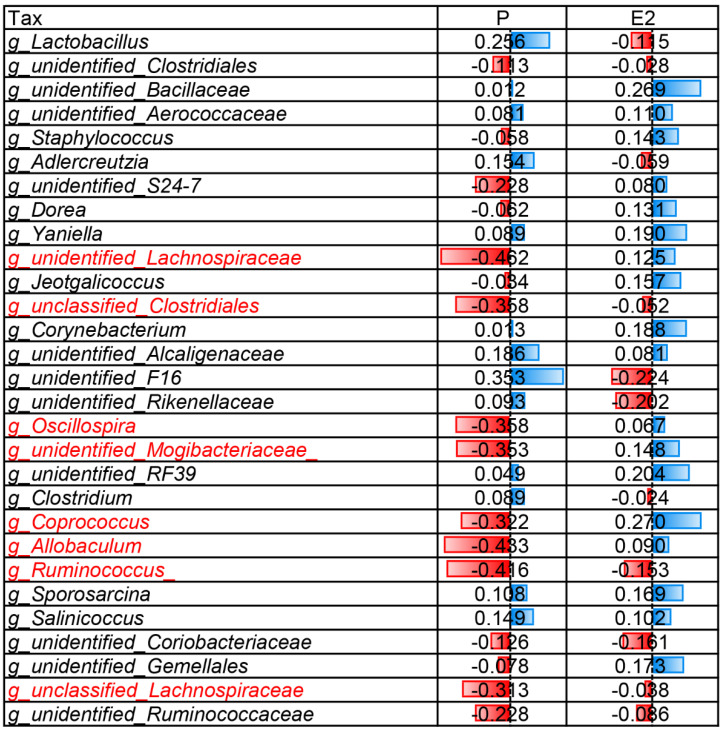
Spearman correlation between serum reproductive hormones and caecal microbiota in mice at the genus level in the different groups.

**Figure 5 vetsci-09-00169-f005:**
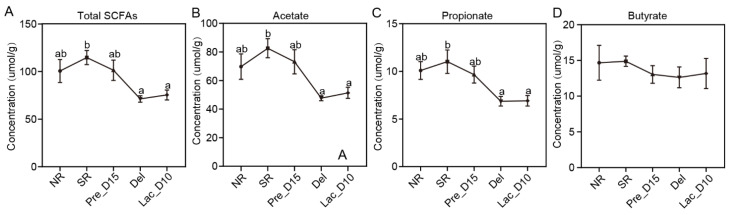
The SCFAs’ concentrations of caecal content. (**A**) The concentration of total SCFAs. (**B**) The concentration of acetate. (**C**) The concentration of propionate. (**D**) The concentration of butyrate. As long as the same letter appears in five groups, the difference between the groups is insignificant; otherwise, the difference between the groups is significant (*p* < 0.05).

**Figure 6 vetsci-09-00169-f006:**
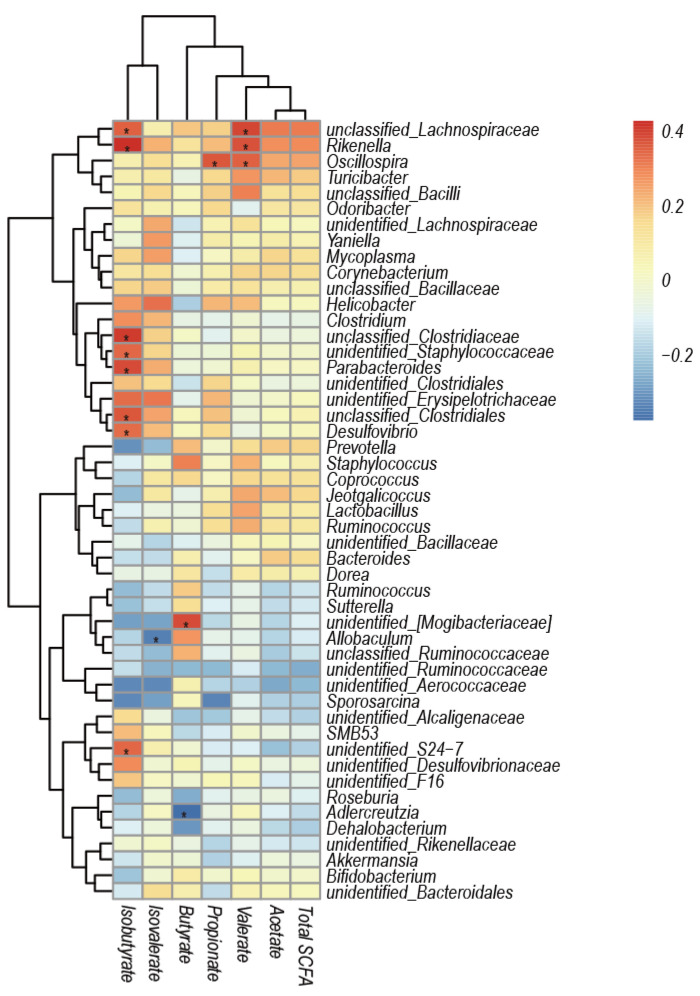
Spearman correlation between SCFAs and caecal microbiota at the genus level. Each spot color in the heatmap corresponds to the ρ of the spearman correlation analysis between microbial abundance and SCFAs. The spot with an asterisk indicates a significant correlation at *p* < 0.05.

## Data Availability

The samples of 16S rRNA gene sequencing were available from the NCBI under accession No. PRJNA744582.

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
