# Peer review of "16S rRNA Gene Sequencing Revealed Changes in Gut Microbiota Composition during Pregnancy and Lactation in Mice Model"

_vetsci, 2022, doi:10.3390/vetsci9040169_

Round 1

Reviewer 1 Report

The study is very interesting and covers the microbiome during pregnancy using mice as models, from which still little is known. The authors used 16S and SCFA data to assess changes during pregnancy, which provides further evidence from a microbial and metabolic perspective. The authors also correlated microbial and SCFA composition in the mice gut.

Major comments:

One main concern is the database used for the taxonomy annotation. According to the Greengenes website, ‘these publicly available versions of the Greengenes database utilize taxonomic terms proposed from phylogenetic methods applied years ago between 2012 and 2013’, which may not be able to accurately identify taxa present in mouse fecal contents. Indeed, this is a caveat of using outdated databases that have been mostly built for human fecal samples. Unfortunately, this is something that needs to be addressed in the manuscript. Indeed, how many reads matched host DNA, what percent of the total reads were microbial and what percent were unassigned?

Minor comments:

Section 2.5 Can the authors specify which variable region was amplified?

Figure 3B NR vs Lac_D10 comparison: The bacterial names are very hard to read because of the font color. Is it possible to change the color to a darker tone?

Author Response

Response to reviewers

We thank the 2 reviewers for their evident care to detail, and comments. Each reviewer offered helpful and constructive suggestions, and as a result we feel that the paper has been significantly improved by their input. For this we are grateful. We have seriously thought about them and provided our response to reviews, for the detailed response, please see below:

Reviewer #1:

The study is very interesting and covers the microbiome during pregnancy using mice as models, from which still little is known. The authors used 16S and SCFA data to assess changes during pregnancy, which provides further evidence from a microbial and metabolic perspective. The authors also correlated microbial and SCFA composition on in the mice gut.

Major comments:

One main concern is the database used for the taxonomy annotation. According to the Greengenes website, ‘these publicly available versions of the Greengenes database utilize taxonomic terms proposed from phylogenetic methods applied years ago between 2012 an d 2013’, which may not be able to accurately identify taxa present in mouse fecal contents. Indeed, this is a caveat of using outdated databases that have been mostly built for human fecal samples. Unfortunately, this is something that needs to be addressed in the manuscript. Indeed, how many reads matched host DNA, what percent of the total reads were microbial and what percent were unassigned?

RE: Good suggestion! The Greengenes database is the most classic 16S species database based on manual sorting. This database has a strong influence on microbial study, and it has a high comparison accuracy. The classification adopts seven levels (Domain, phylum, Class, Family, Genus, Species), which is convenient for understanding and reading. Therefore, our data select Greengenes database for data annotation and alignment. In our data, a total of 2,653,875 high-quality reads from 35 samples with an average of 60,000 clean reads per sample were obtained after quality control, denoising, splicing, and dechimera processing. Using the classify-sklearn algorithm of QIIME2, we found that 5.16% of the data could not be annotated in detail at the phylum level, and the other data have detailed annotation information at the phylum level. The non-annotated data was not included in the subsequent analysis. Regarding host contamination, we used amplicon sequencing (not metagenomic sequencing) in this study, it was not included host DNA information in the high-quality reads after quality control and denoising. The table R1 is the Statistical table of the number of microbial taxa at each level. Based on the above information, we believe that the analysis of the data is reasonable.

Table R1 Statistical table of the number of microbial taxa at each level

(Excludes unidentified and uncultured taxa)

Sample

Phylum

Class

Order

Family

Genus

Species

NR.1

13

28

40

52

53

29

NR.2

7

15

22

40

40

18

NR.3

9

19

29

50

53

26

NR.4

10

17

22

39

39

17

NR.5

10

19

28

46

49

21

NR.6

7

14

18

27

31

13

NR.7

7

12

15

21

25

10

SR.1

8

15

18

35

41

18

SR.2

8

14

21

34

38

14

SR.3

7

15

21

37

42

20

SR.4

12

20

27

53

54

25

SR.5

8

16

22

35

37

21

SR.6

13

27

33

44

40

19

SR.7

10

18

26

43

46

18

Pre_D15.1

9

17

23

35

41

17

Pre_D15.2

6

14

21

33

37

14

Pre_D15.3

8

17

26

54

63

20

Pre_D15.4

8

16

24

46

52

22

Pre_D15.5

10

20

24

35

35

19

Pre_D15.6

10

18

22

35

34

11

Pre_D15.7

8

16

19

30

36

15

Del.1

9

18

27

41

39

17

Del.2

9

16

22

32

37

15

Del.3

7

14

20

34

36

17

Del.4

9

18

26

39

40

19

Del.5

8

15

22

34

36

17

Del.6

8

18

26

48

51

18

Del.7

8

15

22

38

40

15

Lac_D10.1

9

17

27

42

45

23

Lac_D10.2

10

17

22

37

41

20

Lac_D10.3

11

19

29

47

46

22

Lac_D10.4

9

17

25

42

49

19

Lac_D10.5

9

18

26

44

52

21

Lac_D10.6

8

15

24

39

47

28

Lac_D10.7

10

18

26

47

48

21

Minor comments:

Section 2.5 Can the authors specify which variable region was amplified?

RE: Good suggestion! We have revised it. The primers 338F (5’-ACTCCTACGGGAGGCAG CAG-3’) and 806R (5’-GGACTACHVGGGTWTCTAAT-3’) on a thermocycler PCR system (Gene Amp 9700, ABI, USA) were used to amplify the V3-V4 region of the DNA.

Figure 3B NR vs Lac_D10 comparison: The bacterial names are very hard to read because of the font color. Is it possible to change the color to a darker tone?

RE: Good point! We have revised it. 

Reviewer 2 Report

This is a work about the gut microbiota characterization and metabolic inducing changes in mice, at sexually receptive time and during pregnancy and lactation. The authors found significative differences in gut microbes and metabolites, as modulators of different metabolic pathways, and suggest some key microbes as potential biomarkers for future probiotic applications.

General concept comments:

The manuscript is well organized and clear, with good number of recent references. The experimental design is appropriated. The authors respond to the questions raised, and the results encourage to more studies.

Specific comments:

MATERIALS AND METHODS

- Mouse estrus cycle used for sample collection needs a little correction/clarification, since sexually-receptive and non-receptive stages in the text are not the same than in the figure 1 and in the legend.

- line 143, mistaken reference for primers

RESULTS

- Since the mice that did not become pregnant (Non-receptive) also gained a considerable weight (over nourishment?), could the diet have affected the gut microbiota to some extent? Gut microbiota composition and weight are linked, as referred in reference 36. A comment about that would be helpful.

REFERENCES

The References section needs a general review of the formatting. Some examples are:

Repeated reference 39 and 40, probably mistaken by the original article from where primers information come from; distinguish between two references with the same 1st author name and year, such as 30 - 31 and 57 – 58; Reference 34, does not follow the required format regarding the authors; Missing pagination such as nº33, 46…; The names of the journals, some in uppercase and others in lowercase.

Minor comments

- In the Introduction, lines 64-65. “…In women and mice, the gut microbial composition in the third trimester of pregnancy…”, I suggest to change “third trimester” by alternative expression (i.e.: the last third or late pregnancy…), since for mice is a shorter period.

- In the Material and Methods: Thermocycler brand and model is missing

- In the Results, line 246, between sentences, it is probably a comma and not a dot, for the sentence to make sense.

Author Response

Response to reviewers

We thank the 2 reviewers for their evident care to detail, and comments. Each reviewer offered helpful and constructive suggestions, and as a result we feel that the paper has been significantly improved by their input. For this we are grateful. We have seriously thought about them and provided our response to reviews, for the detailed response, please see below:

Reviewer #2:

This is a work about the gut microbiota characterization and metabolic inducing changes in mice, at sexually receptive time and during pregnancy and lactation. The authors found significative differences in gut microbes and metabolites, as modulators of different metabolic pathways, and suggest some key microbes as potential biomarkers for future probiotic applications.

RE: Thanks for your positive comment.

General concept comments:

The manuscript is well organized and clear, with good number of recent references. The experimental design is appropriated. The authors respond to the questions raised, and the results encourage to more studies.

RE: Thanks for your positive comment.

Specific comments:

MATERIALS AND METHODS

- Mouse estrus cycle used for sample collection needs a little correction/clarification, since sexually-receptive and non-receptive stages in the text are not the same than in the figure 1 and in the legend.

RE: Done. The caecal contents and mice serum were collected during the non-receptive (NR) stages (proestrus and estrus; n=7), sexually-receptive (SR) stages (diestrus and metestrus; n=7).

- line 143, mistaken reference for primers

RE: Done.

RESULTS

- Since the mice that did not become pregnant (Non-receptive) also gained a considerable weight (over nourishment?), could the diet have affected the gut microbiota to some extent? Gut microbiota composition and weight are linked, as referred in reference 36. A comment about that would be helpful.

RE: Good point! In the Materials and methods section, we detail explained that the diet of all mice did not change during the whole trial period. Pls see L93. In Figure 2A, we showed the initial body weight and the sampled body weight (Final) of the mice at each stage. The bodyweight of mice prior to estrus identification was defined as initial body weight, and there was no significant difference between groups (Figure 2A). After two estrous cycles in the breeding experiment, we sampled the mice in the NR group of mice, so the weight gain of the mice was related to age. We present this in the Results section of the manuscript. In addition, weight gain in gestational mice is associated with fetal and maternal obesity during the gestation period, and we have previously listed causal studies of gut microbiota and gestational obesity in the Discussion section.

REFERENCES

The References section needs a general review of the formatting. Some examples are:

Repeated reference 39 and 40, probably mistaken by the original article from where primers information come from; distinguish between two references with the same 1st author name and year, such as 30 - 31 and 57 – 58; Reference 34, does not follow the required format regarding the authors; Missing pagination such as nº33, 46…; The names of the journals, some in uppercase and others in lowercase.

RE: Done.

Minor comments

- In the Introduction, lines 64-65. “…In women and mice, the gut microbial composition in the third trimester of pregnancy…”, I suggest to change “third trimester” by alternative expression (i.e.: the last third or late pregnancy…), since for mice is a shorter period.

RE: Good point! Done.

- In the Material and Methods: Thermocycler brand and model is missing

RE: Done.

- In the Results, line 246, between sentences, it is probably a comma and not a dot, for the sentence to make sense.

RE: Done.

Round 2

Reviewer 1 Report

The authors seemed to have addressed the questions

Author Response

RE: Thanks for your positive comment.